



# Carbon dioxide variations in the upper troposphere and lower stratosphere from GOSAT TANSO–FTS TIR profile data

Akihiro Honda[1], Nawo Eguchi[2], and Naoko Saitoh[3]

[1]Interdisciplinary Graduate School of Engineering and Sciences, Kyushu University, Kasuga Park 6-1, Kasuga, Fukuoka, Japan; now at Mitsubishi Heavy Industries Ltd.
[2]Research Institute for Applied Mechanics (RIAM), Kyushu University, Kasuga Park 6-1, Kasuga, Fukuoka, Japan
[3]Center for Environmental Remote Sensing, Chiba University, Yayoi-cho 1-33, Inage-ku, Chiba, Japan

**Correspondence:** Nawo Eguchi (nawo@riam.kyushu-u.ac.jp)

**Abstract.** Carbon dioxide ($CO_2$) variations in the upper troposphere and lower stratosphere (UT and LS; pressure = 300–70 hPa) were investigated with profile data derived from the thermal infrared region (Band 4: 5.5–14.3 $\mu$m) of the Thermal And Near-infrared Sensor for carbon Observation (TANSO)–Fourier Transform Spectrometer (FTS) onboard the Greenhouse gas Observing SATellite (GOSAT). The vertical profile data (Level 2; version 1) of $CO_2$ mixing ratios were obtained by bias correction using the *in situ* aircraft data. Data with sufficient sensitivity were used for the analysis by selecting values with higher degrees of freedom. The analysis period is over four years from January 2010 to December 2013.

The increase of $CO_2$ mixing ratios obtained with the bias correction in the UT and LS was 1.8–2.4 ppmv year$^{-1}$ from 2010–2013, which is consistent with a previous observational study and almost the same as for the lower troposphere (LT). The seasonal variations in the UT and LS show that a maximum peak exists one or two months after the peak in the LT, and has an amplitude of 4–5 ppmv, which is less than that in the LT. The inter-annual variations observed in the tropical UT appear to be affected by ENSO events; higher (lower) $CO_2$ mixing ratios were observed during La Niña (normal/El Niño) periods. Intra-seasonal variations over the Asian summer monsoon region were associated with both vertical and horizontal motions owing to deep convection and monsoonal anticyclonic circulation, respectively.

## 1 Introduction

The Intergovernmental Panel on Climate Change (IPCC) concluded that the main cause of global warming since the middle twentieth century is highly likely to be human activity (IPCC, 2013, 2021). Carbon dioxide ($CO_2$) is the major anthropogenic greenhouse gas causing this warming, and it has been reported that the mixing ratios of $CO_2$ in the troposphere has risen from 280 to 400 parts per million volume (ppmv) in the past 170 years. The $CO_2$ mixing ratios in 2018 and 2019 were 407.8 and 410.5 ppmv, respectively (WMO Greenhouse Gas Bulletin, 2019, 2020), and an annual record was set in 2020 of 413.2 ppmv (WMO Greenhouse Gas Bulletin, 2021), which was expected despite the reduction in economic activity due to COVID-19. It has been reported that the stratosphere is cooling (IPCC, 2013, 2021), which could be caused by increasing $CO_2$ mixing ratios in the stratosphere in addition to a decrease in stratospheric ozone (cf. Langesmatz et al., 2003; Ramaswamy et al., 2006), because $CO_2$ warms the troposphere and cools the stratosphere (Wang et al., 2020). Apart from $CO_2$ production due





to methane oxidation in the stratosphere, which is up to 1 ppmv (Boucher et al., 2009), there is no source of $CO_2$ above the
stratosphere other than the troposphere. It is clear that the increase in the stratospheric $CO_2$ content is caused by $CO_2$ being
derived from the troposphere.

The rise in $CO_2$ amount in the stratosphere is contributing to climate change (e.g., by providing suitable conditions for
ozone depletion), and therefore warrants further investigation. $CO_2$ is a long-lived gas and it is often assumed to have a
constant mixing ratio in general climate models because it is generally thought to be well-mixed in the atmosphere. However,
observations have revealed that the spatial and temporal distributions of $CO_2$ mixing ratios are variable (e.g., Sawa et al., 2008),
thus monitoring the global amount of $CO_2$, including in the stratosphere, is essential for robust climate change predictions.
The stratospheric $CO_2$ amount on a global scale and exchange processes between the upper troposphere and lower stratosphere
(UT and LS; 300–70 hPa) are not well understood. Therefore, the details of inflow of $CO_2$ from the UT to LS remain unclear.

Aircraft and ground-based balloon observations have obtained *in situ* measurements of greenhouse gases in the UT and
stratosphere (e.g., Nakazawa et al., 1995; Aoki et al. , 2003; Sawa et al., 2008; Umezawa et al., 2018). These previous studies
were based on *in situ* observations with high accuracy, and found year-to-year variations, increases, and seasonal variations in
$CO_2$ mixing ratios in the UT–LS region. However, the spatial resolution and frequency of these *in situ* observations are limited.
For example, the horizontal and vertical regions were limited to aircraft routes and pressure levels (i.e., mainly the UT–LS), and
the ground-based observations have limited spatial coverage and temporal resolution. Furthermore, the intra-seasonal variation
of $CO_2$ in the UT–LS has not yet been investigated. Therefore, in order to understand the effect of stratospheric $CO_2$ on climate
change, it is necessary to determine the $CO_2$ variations in the UT–LS region on a global scale.

Stratosphere–troposphere exchange (STE) normally occurs over the tropical regions, especially the western Pacific (i.e.,
the stratospheric fountain; (e.g., Newell and Gould-Stewart , 1981; Holton et al., 1995)) in the boreal winter, and the monsoon
region in the boreal summer in Asia and North America (e.g., Gettelman et al., 2004; Santee et al., 2017; Yu et al., 2020). The
intrusion process in the tropics is closely related to the strength of the Brewer–Dobson circulation (BDC), which is the general
meridional circulation in the stratosphere driven by planetary waves propagated from the troposphere (Andrews et al., 1987).
The BDC is stronger in the boreal winter than in the austral winter, which means the intrusion process has a seasonal cycle
(e.g., Mote et al., 1996). In addition to the strength of the BDC, there are perturbations originating from the troposphere on
hourly to interannual timescales, related to waves, convection, and radiation. For example, overshooting convective clouds
cause tropospheric air to intrude directly into the stratosphere (e.g., Eguchi et al., 2016). Over the summer monsoon region, the
stratosphere–troposphere exchange process is active due to the active convection and related wind fields (i.e., the Tibetan anti-
cyclonic circulation) on intra-seasonal scales (e.g., Randel and Park, 2006). Over these areas, there are less *in situ* observations
of the UT–LS region, and there are prominent intra-seasonal phenomena that characterize the area, such as the Madden–Julian
Oscillation (MJO) (30 – 60 days) (Madden and Juluan, 1994) and the boreal summer intra-seasonal oscillation (10 – 90 days)
(e.g., Kang et al., 1999; Annamalai and Slingo, 2001).

The $CO_2$ mixing ratios obtained from space–born measurements were determined by the SCanning Imaging Absorption
spectroMeter for Atmospheric CHartographY (SCIAMACHY) (Buchwitz et al., 2007), the Atmospheric Chemistry Experiment-
Fourier Transform Spectrometer (ACE-FTS) (Foucher et al., 2011), The Tropospheric Emission Spectrometer (TES) (Kulawik et al.,





2010), The Atmospheric Infrared Sounder (AIRS) (Maddy et al., 2008), and the Infrared Atmospheric Sounding Interferome-

ter (IASI) (Crevoisier et al., 2009). However, the $CO_2$ data from these satellite measurements are insufficient to investigate the $CO_2$ variations in the UTLS, particularly on intra-seasonal timescales. The Greenhouse gas Observing SATellite (GOSAT) was launched in January 2009 to dedicated to observations of the mixing ratios of greenhouse gases such as $CO_2$, $CH_4$, and $H_2O$ on a global scale (Yokota et al., 2009). The Thermal And Near-infrared Sensor for the carbon Observation–Fourier Transform Spectrometer (TANSO–FTS) instrument (e.g., Kuze et al., 2016) onboard GOSAT measures the greenhouse gas mixing ratios

from the surface to the LS using the thermal infrared band. GOSAT observes every three days over almost the entire region from 80°N to 80°S in a polar orbit with a 100 min cycle. Ground-based and aircraft instruments made 248 observations in September 2019 (WMO WDCGG Data Summary, 2020), but about 80,000 observations of the same quality, both with and without cloud cover, are made by GOSAT each month. This enables data to be acquired for areas that cannot be studied by ground-based observations.

GOSAT has been observing from June 2009 to the present day and GOSAT-2, which was launched on 29 October 2018, also monitors greenhouse gases from space. It is now possible to determine the three-dimensional distribution of greenhouse gases, and also intra-seasonal (30–60 days), seasonal, and inter-annual (a few years) variations since June 2009. These data are useful for numerical models and improving the accuracy of emission measurements. For example, the total column $CO_2$ mole fraction ($XCO_2$) data derived from shortwave infrared (SWIR) GOSAT observations are used for estimating the surface flux with an

atmospheric transport model (e.g., Maksyutov et al., 2021; Patra et al., 2021; Houweling et al., 2015; Takagi et al., 2011).

Numerous previous studies have been conducted on the total atmospheric column $CO_2$ mole fraction in the middle (MT) and lower (LT) troposphere using satellite observations. The seasonal variations were investigated by Yoshida et al. (2013); Lindqvist et al. (2015). Ying et al. (2019) determined inter-annual and seasonal variations of $CO_2$ from GOSAT and the AIRS onboard the Aqua satellite, and Jiang et al. (2010) measured MT $CO_2$ variations using the AIRS. However, there have been

fewer studies of $CO_2$ variations in the UT–LS on a global-scale by satellite measurements. Diallo et al. (2017) investigated variations in $CO_2$ in the UT and stratosphere using a Lagrangian backward trajectory model driven by ERA-Interim reanalysis meteorology and tropospheric $CO_2$ measurements.

In this study, we investigated $CO_2$ mixing ratios in the UT–LS region on a global scale at several temporal scales, from intra-seasonal, seasonal, to inter-annual, using GOSAT profile data after corrections for data bias and quality. Section 2 describes the

$CO_2$ profile data derived from the thermal infrared band of the GOSAT TANSO–FTS, and the bias correction and analytical methods, including the definition of the temporal variations. Section 3 presents the $CO_2$ trend from 2010–2013, and seasonal, inter-annual, and intra-seasonal variations in the UT–LS. Finally, Section 4 summarises our results and conclusions.

## 2  Data and methodology

### 2.1  GOSAT observation overview and spectral information

The GOSAT satellite consists of two sensors: The TANSO–FTS and the TANSO–Cloud and Aerosol Imager (TANSO–CAI). Bands 1, 2, and 3 of the TANSO–FTS sensor observed the short infrared wavelength (SWIR), with wavelengths of 0.76, 1.6,





and 2.0 $\mu$m, respectively, which are the spectra of solar radiation reflected from the ground surface (e.g., Kuze et al., 2012, 2016; Yoshida et al., 2011). From the SWIR band, the column-average mole fractions (XCO$_2$, XCH$_4$, and XH$_2$O) of CO$_2$, CH$_4$, and H$_2$O can be retrieved (Yoshida et al., 2011, 2013). Band 4 of the TANSO–FTS is the TIR band, and observes the

infrared spectrum (5.5–14.3 $\mu$m) of Earth's radiation emitted from the atmosphere and the surface. From the TIR band, the vertical mixing ratio distributions of CO$_2$ and CH$_4$ can be retrieved (Saitoh et al., 2009, 2016). Given that the resolution of the spectra is 0.2 cm$^{-1}$ higher than that of conventional satellite equipment, except for band 1, the atmospheric mixing ratio data are of high accuracy. For example, the accuracy is about 4 ppmv for XCO$_2$ (Yoshida et al., 2011). Moreover, the high-resolution spectra allow high-resolution vertical data to be obtained by TIR.

The TANSO–CAI sensor observes clouds, aerosols, and surface conditions during the day. From the TANSO–CAI observational data, it can be determined whether there is cloud over a wide range, including the field of view (instant field of view, IFOV, which is approximately 10.5 km in diameter) of the TANSO–FTS sensor (Ishida and Nakajima, 2009; Ishida et al., 2011). If there is an aerosol or a cloud within the IFOV, then the retrieval processing of CO$_2$, CH$_4$, and H$_2$O mixing ratios is not conducted on the SWIR and TIR spectra. However, optically thinner clouds are not detected by the TANSO–CAI; therefore,

the accuracy of the TIR profile data has a bias with respect to cloud frequency, especially for thinner cloud.

This study used Level 2 vertical profile data (version 1; latest version) retrieved from the GOSAT TIR spectrum acquired over the clear sky for both day and night for four years from 1 January 2010 to 31 December 2013. The retrieval process for the vertical mixing ratio distribution of CO$_2$ is explained in detail in Saitoh et al. (2009, 2016). The pressure levels are 28 layers from the surface to 0.1 hPa. The present study defined the UT–LS region as 287.30–90.85 hPa, and the number of retrieved

layers is 9 to 14 in Table 1 of Saitoh et al. (2016). The CO$_2$ mixing ratios data were averaged in a 5°box for the longitude and latitude. The algorithm was described in detail by Saitoh et al. (2016), and the altitude distribution data for 28 layers were specifically derived. The algorithm has a sensitivity peak in the upper troposphere.

## 2.2 Bias correction and data pre-processing

Saitoh et al. (2016) determined the TIR CO$_2$ profile in the UT–LS region using Continuous Measurement Equipment (CME)
observations of the Comprehensive Observation Network for TRace gases by AIrLiner (CONTRAIL) (Machida et al., 2008), and the accuracy of the CO$_2$ measurements was less than 0.2 ppmv (Machida et al., 2008, 2011). It was found that the CO$_2$ mixing ratios in the UT–LS had a maximum bias of 2.3 ppmv between the equator and 40°N in the boreal summer and 2.4 ppmv between 20°N and 40°N in the boreal spring during the four years from 2010 to 2013. In summary, Saitoh et al. (2016) found that the CO$_2$ profile data had consistent negative biases of 1.0%–1.5% as compared with the CME CO$_2$ data in the LT and

MT regions (736–287 hPa). Saitoh et al. (2017) found that the CO$_2$ profile data at the CME altitude range (approximately 11 km) was less than 0.5 ppmv at lower latitudes and 1.0 ppmv at middle and high latitudes. In addition, after applying the bias correction coefficient based on a comparison with CONTRAIL, both the trends and seasonal fluctuations showed good agreement with the Nonhydrostatic Icosahedral Atmospheric Model (NICAM)–Transport Model (TM) (Niwa et al., 2011, 2012) after inverse analysis. The present study conducted a bias correction at the retrieved layers (287.30–90.85 hPa) for each month



at 10° latitude intervals from 30°S to 70°N, and determined the pressure level with reference to the bias-corrected data of Saitoh et al. (2016, 2017).

In addition, data with a higher degree of freedom (DOF), with more than two times the standard deviation from the average of each month, were used for the analysis, because we intended to use data that were not constrained by a priori information. Furthermore, anomalous data that were more than five times the standard deviation from the average of each month were not 130 used for the analysis.

The trend was derived from the slope of a linear least-square fit defined by equation (1) from the monthly mean time-series of $CO_2$ mixing ratios from January 2010 to December 2013.

$$CO_{2t}(t) = at + b \qquad (1)$$

, where $a$ and $b$ are the slope and intercept, respectively. The trend was calculated at each pressure level and latitude at 135 5°intervals. Because the unit of $a$ is ppmv month$^{-1}$, the trend is $a$ multiplied by 12 months.

The seasonal cycle was derived from a harmonic analysis (Fourier time-series analysis), similar to Nakazawa et al. (1991), of the monthly mean time-series subtracted from the linear trend defined by equation (1). Seasonal cycle components were extracted from the observed time-series data:

$$S(t) = \sum_{i=1}^{k} (A_i \sin 2\pi it + B_i \cos 2\pi it) \qquad (2)$$

, where $A_i$ and $B_i$ are the amplitude parameters of the sine and cosine terms, respectively. $t$ is the elapsed time since January 2010. $i$ is the order and $k$ was set to three, which satisfactorily expresses the sub-seasonal and annual cycles.

For analysis of the inter-annual $CO_2$ variations, we used the anomaly data from the four-year monthly mean derived from equation (2), after subtracting the trend for each month and pressure and latitude.

Meteorological data, including the temperature and horizontal wind field, were obtained by ECMWF reanalysis version 5 145 (ERA5) ().

## 3 Results

### 3.1 $CO_2$ mixing ratio trends from 2010 to 2013

Figure 1 shows the trend of $CO_2$ mixing ratios at 250 hPa. The trend at 250 hPa was obtained from bias-corrected data within the latitudinal range of 30°S–70°N. The filled circles in Fig.1 show data from latitudes where the data were bias-corrected. 150 The trend at 250 hPa was around 2.0 ppmv year$^{-1}$.

Table 1 shows a summary of the trend at several pressure levels during the analysis period. The trend was 1.82–2.08 ppmv year$^{-1}$ in the UT (150 and 250 hPa) and the LS (100 hPa), and 1.87 ppmv year$^{-1}$ at 500 hPa. The trend in the Southern Hemisphere (30°S–Eq.) was relatively larger than in the Northern Hemisphere (Eq.–30°N); the difference at 250 hPa was 0.15 ppmv year$^{-1}$. In particular, the trend was large over the Atlantic, Indian, and western Pacific oceans (not shown). These values





are consistent with *in situ* observations, such as CONTRAIL, which obtained an increase of 1.8–2.0 ppmv/yr from 2005–2010 (Sawa et al., 2012). For example, the growth rate at Mauna Loa was 2.22 ppmv year$^{-1}$ during the analysis period (c.f. Thoning et al., 1989). The increase of 1.82–2.08 ppmv/yr in the UT–LS is similar to that near the surface (2.0–2.9 ppmv/yr) (WMO Greenhouse Gas Bulletin, 2012, 2013, 2014), and thus it appears that $CO_2$ is well mixed from the LT to UT.

### 3.2   Seasonal cycle

Figure 2 shows a time–latitude cross-section of zonal mean and four-year averaged $CO_2$ mixing ratios at each pressure level from January 2010 to December 2013. The four-year average values were derived from the de-trended time-series at each latitude and pressure.

At 500 hPa (Fig.2d), a high $CO_2$ mixing ratio appears at higher latitudes between 40°N and 65°N in March and April. Prior to that, $CO_2$ mixing ratios increase from November. In the Northern Hemisphere, the peak month of the seasonal variation

becomes later approaching the equator. At low latitudes in the Southern Hemisphere (around 30°S), the peak is in September. The seasonal amplitude is approximately 8 ppmv at higher latitudes and this decreases at lower latitudes. The peak of the seasonal cycle around the equator is June. These features are similar to that at the near-surface, which reflects seasonal controls on vegetation in the Northern Hemisphere (e.g. Keeling et al., 1996).

At 250 hPa, the maximum $CO_2$ mixing ratios occur in May at lower latitudes (30°S to 30°N), which is one or two months

earlier than at mid-latitudes at 500 hPa. The $CO_2$ mixing ratio increases in January or February and decreases from June. The seasonal amplitude at lower latitudes is approximately 5 ppmv, which is smaller than at lower altitudes. Furthermore, the amplitude at higher latitudes (north of 30°N) is smaller than that at lower latitudes.

At 150 hPa, there are two maximum $CO_2$ mixing ratio peaks at 30°N from April to May and at 20°S from May to July. The southern peak at 20°N extends southward with each month. The northern peak at 30°N is independent of the southern peak

and extends northward, but only until June. Similar to 250 hPa, the $CO_2$ mixing ratios at lower latitudes were higher than those at higher latitudes.

At 100 hPa, the $CO_2$ mixing ratio peak at 15°N occurred in May and June. The higher mixing ratio extended to the Southern Hemisphere across the equator, and it appears that this extension causes the seasonal cycle at lower latitudes in the Southern Hemisphere. The amplitude was approximately 4 ppmv. In summary, a seasonal $CO_2$ cycle was observed in the UT, but the

amplitude was small. The peak was located at the lower latitudes and occurred earlier than the peak at the lower altitudes (150 and 250 hPa).

Figure 3 shows the latitude and pressure sections of zonal and monthly averaged $CO_2$ mixing ratios from 2010 to 2013 after de-trending. The pressure range is 735–30 hPa. Solid and dashed contours show negative and positive pressure vertical velocities ($-/+1.0$ Pa/s) and dotted contours show the potential temperatures at 370 K. The 370 K potential temperature

defines the physical surface of the tropopause.

Seasonal variation is clearly evident at mid- and high latitudes in the Northern Hemisphere, especially below the MT (300 hPa). High mixing ratios of $CO_2$ occur at mid- and high latitudes and extend from the south up to the equator between March and May. After June, the $CO_2$ mixing ratios at mid- and high latitudes decreased until September, and then increased gradually





from October to the following May. At lower latitudes, high $CO_2$ mixing ratios from the LT to UT were observed from April
to June, which are related to the location of upward motion in a Hadley cell in the Northern Hemisphere (Fig. 3).

There are high $CO_2$ mixing ratios in the UT in the equatorial region, which is the Tropical Tropopause Layer (TTL); the
pressure range is 150–70 hPa. This high $CO_2$ mixing ratio region is isolated from the MT throughout the year, except for during
March–June, when high $CO_2$ mixing ratios extend from the LT to UT and lower latitudes (equator to 30°N) due to upward
vertical motion, such as Hadley cells. This feature is consistent with the seasonal cycle in the UT (250 hPa) shown in Figure
2c.

During the boreal winter (austral summer) season (December–February), the lowest mixing ratio was observed up to 200
hPa along the upward vertical velocity zone located to the south of the equator, which can reach a higher altitude than during
the northern summer. During the boreal summer when the annual cycle of $CO_2$ is at a minimum, the upward motion was active
over the Asian and American monsoon regions (15°N–45°N), where the $CO_2$ mixing ratio gradually decreased in the MT and
UT.

Given that the source of $CO_2$ , particularly anthropogenic $CO_2$ , is at lower altitudes at mid-latitudes in the Northern Hemi-
sphere, the high $CO_2$ mixing ratio is transported from mid-latitudes to lower latitudes in the LT by dynamical circulation, such
as Ferrel and Hadley circulations. This suggests that the high $CO_2$ mixing ratio is transported from lower altitudes during the
boreal spring to early summer, and remains in the equatorial region where there are no anthropogenic sources or sink of $CO_2$ .
Furthermore, the $CO_2$ mixing ratios in the equatorial region of the UT reached a minimum around September, which means
that the aforementioned transport from mid-latitudes to lower latitudes by general circulation, such as Ferrel and Hadley circu-
lations, takes 1–2 months. In the LS above the tropopause at approximately 350 K, the $CO_2$ mixing ratio decreased gradually
with increasing altitude. In the Northern Hemisphere, the $CO_2$ mixing ratio vertical gradient varied with season. The gradient
in the Southern Hemisphere was steeper than in the Northern Hemisphere and, for example, the $CO_2$ mixing ratio decreased
from 385 ppmv at 250 hPa to 370 (375) ppmv at 100 hPa at 60°S (60°N) in January. This suggests that the $CO_2$ mixing ratio
in the UT and LS depends on vertical and horizontal transport in the troposphere.

### 3.3 Year-to-year variation

Figure 4 shows a time–latitude plot of $CO_2$ mixing ratios at 100, 150, 250, and 500 hPa from January 2010 to December 2013,
using de-trended data at each latitude.

At 500 hPa, the seasonal variation at northern high latitudes (50°N) is large, as shown in Fig. 2. The highest $CO_2$ mixing
ratio was observed in April at high latitudes and extended to lower latitudes in each year. In April 2013, this extension was
larger than for the other years. Note that the bias correction was not applied to the data at high latitudes and, therefore, the
bias remains at high latitudes. The minimum $CO_2$ mixing ratios were located at latitudes higher than 60°N and at around 15°N
from June to November. The southern minimum extended to the Southern Hemisphere from September to April. In 2011, the
minimum $CO_2$ mixing ratio near the equator was larger than for the other years.

At 250 hPa, the year-to-year variations appear to be larger than at other pressures. The 250 hPa pressure level at lower
latitudes is in the UT and, in general, corresponds to the maximum of zonal winds that are influenced by deep convection and





general circulation. At mid- and high latitudes, the 250 hPa pressure level is located in the LS. At lower latitudes (30°N/S), one peak of high $CO_2$ mixing ratios was observed in 2010 and 2013, and two peaks were observed in 2011 and 2012. These

peaks at lower latitudes reflect upward motion associated with deep convection and Hadley circulation. The years 2011 and 2012 were normal years in terms of ENSO and, therefore, the upward motion was weaker than in La Niña and El Niño years (e.g. Oort and Yienger, 1996; Yun et al., 2021). The high $CO_2$ mixing ratios extended to higher latitudes in both hemispheres, however, the northern maximum extended further than the southern maximum from April to June.

At 150 hPa, two maximum $CO_2$ mixing ratios were located at 35°N and the lower latitudes of 15°N/S. The southern

maximum shows larger year-to-year variations than the northern maximum. The peak at 35°N occurred in April, which suggests that the high mixing ratio at 150 hPa could have resulted from transport from the LT at mid-latitudes. The 150 hPa pressure level corresponds to the base of the TTL at lower latitudes, and the deep convection zone in the tropics.

At 100 hPa, the higher mixing ratios are located at 15°N in May–June each year and extend to the Southern Hemisphere in June, and also the mid-latitudes in the Northern Hemisphere. The extension to higher latitudes was more prominent in the

Northern Hemisphere than in the Southern Hemisphere. The 100 hPa pressure level in the tropics is in the TTL (within 25°N/S), but is located in the LS in extratropical regions. This suggests that high $CO_2$ mixing ratios gradually enter the LS at 100 hPa during the boreal spring and early summer, and the seasonal cycle is then influenced by dynamic processes (i.e., vertical transport due to deep convection associated with monsoons), because $CO_2$ is a long-lived species in the UT and stratosphere.

Figure 5 is similar to Figure 4, but the anomaly shown is the seasonal cycle at 150 and 500 hPa. In the tropics at 150 hPa

(Fig. 5(a)), positive anomalies occurred from April to February in 2010/2011 and from January to December in 2013. However, a negative anomaly occurred from June 2011 to December 2012. The positive (negative) anomalies correspond to La Niña (El Niño or normal) phases. The positive anomaly extended over the tropics, and the negative anomaly was located on both sides of the positive anomaly in the boreal summer of 2010.

The same features are evident at 250 hPa. The zonal averaged amplitude of the year-to-year variation was ±1.25 ppmv at 250

hPa at the equator, and the amplitude at 150 hPa was slightly larger than that at 250 hPa. At 100 hPa, year-to-year variations were not clear, although weak variations were observed in the northern subtropics. The opposite features were observed in the tropics at 500hPa (Fig. 5(b)); the negative (positive) anomalies were in La Niña (El Niño or normal) phases, which is consistent with a previous study (e.g. Jiang et al., 2010).

Given that the zonal averaged vertical wind in the tropics was weaker during the La Niña period than in normal years and El

Niño periods (not shown), it is possible that air with higher $CO_2$ mixing ratios preferred to remain in the MT and, as a result, there was less air transport to the UT–LS during the La Niña period. This again shows that the $CO_2$ mixing ratio in the UT–LS depends on the magnitude of upward motion.

### 3.4 Intra-seasonal variation

The Asian summer monsoon region is an important area of stratospheric and tropospheric (ST) exchange processes. In partic-

ular, the intra-seasonal variations over this region are active and reflect the seasonal effects of the Asian monsoon from June to September. Xiong et al. (2009) found that the strong transport of atmospheric pollutants from LT to UT in Asian during the



monsoon season by AIRS CH$_4$ observation. The intra-seasonal variations of trace gases have been investigated to understand ST processes (e.g. Park et al., 2007, 2008, 2009). Intra-seasonal variations in carbon monoxide (CO) as a tropospheric tracer and ozone and HCN as a stratospheric tracer both reflect the effects of vertical transport via deep convection and horizontal

advection caused by anticyclones associated with the Asian monsoon.

The intra-seasonal variations of CO$_2$ mixing ratios varied from year-to-year, especially during the boreal spring to autumn seasons. Figure 6 shows the time–latitude section of CO$_2$ mixing ratios at 250 hPa over the Asian summer monsoon region averaged between 60°E and 120°E from the equator to 45°N from 1 April to 31 October of each year. The data were de-trended and are a seven-day running average.

There are CO$_2$ mixing ratio variations over durations of a few weeks in the UT, and high mixing ratios were observed from May to June, especially in May 2010 and May–June in 2013. Given that the high CO$_2$ mixing ratios were transported from the LT by deep convection, as evident from the seasonal variation, it appears that the intra-seasonal variation was also related to deep convection. After June, the CO$_2$ mixing ratios started to decrease gradually from high latitudes, except in August 2012.

Figure 7 shows a horizontal section of CO$_2$ mixing ratios at 250 hPa from 16 June to 1 September 2012, with 15-day intervals

obtained from the 14-day mean. The high mixing ratios of CO$_2$ were mainly located on the southwestern side of the Tibetan Plateau high (i.e., the Asian monsoon high) in June (Fig.7(a)). The high mixing ratio in the UT was caused by vertical transport due to deep convection associated with the Asian Monsoon. The maximum mixing ratio area moved northward of the Tibetan Plateau high in July when the Tibetan Plateau high extended to the east and strengthened (Fig.7(b,c)). At the same time, the CO$_2$ mixing ratio on the eastern side of the study area decreased gradually. There is an east–west change in mixing ratio.

In August, the high-mixing ratio area (>393 ppmv) was diminished, and lower CO$_2$ mixing ratios were transported horizontally from the northeastern part of the study area to lower latitudes along the Asian monsoon high. In September when the Tibetan Plateau high weakened, the CO$_2$ mixing ratio recovered in the western part of the study area. CO$_2$ mixing ratios on an intra-seasonal timescale are influenced by vertical transport due to deep convection and horizontal advection caused by horizontal wind fields in the UT.

**4   Summary and conclusions**

CO$_2$ mixing ratio variations in the UT and LS were investigated with the GOSAT TANSO-FTS TIR Level 2 dataset (version 1) on intra-seasonal to inter-annual timescales. The data were bias-corrected based on Saitoh et al. (2016, 2017) and excluded smaller DOF values. The level 2 data were converted to a 5°N box grid and monthly mean values, except for the intra-seasonal analysis.

The increase of CO$_2$ in the UT–LS was 1.8–2.4 ppmv year$^{-1}$, which is consistent with that obtained from *in situ* aircraft-obtained data by CONTRIAL (e.g., Machida et al., 2008). The increase in the tropical Southern Hemisphere in the MT and UT (250 and 500 hPa) was slightly (approximately 0.5 ppmv year$^{-1}$) larger than that in the Northern Hemisphere.

In terms of the seasonal cycle, the maximum CO$_2$ mixing ratio was in May–June in northern mid-latitudes, and one or two months later than in the LT and MT. This suggests that vertical transport and mixing between the LT and UT occurs on a

timescale of one or two months. The amplitude of the seasonal cycle in the UT was 4–6 ppmv, which is ≤50% of that in the LT and MT. This is due to the absolute mixing ratio decreasing with altitude and, to a lesser extent, mixing with a low $CO_2$ mixing ratio air mass.

Inter-annual variations were clearly evident in the tropics, and appear to be affected by ENSO events. At 150 hPa, higher (lower) $CO_2$ mixing ratios occurred during La Niña (normal/El Niño) periods. This suggests that the higher $CO_2$ mixing ratios 295   in the UT during the La Niña periods were induced by weaker vertical transport than during normal/El Niño periods.

In terms of intra-seasonal variations, $CO_2$ mixing ratios over the Asian summer monsoon region were variable on a timescale of a few weeks, which reflect vertical and horizontal transport owing to deep convection and monsoonal anticyclonic circulation, respectively. Vertical transport increased the $CO_2$ mixing ratios in the upper layer for short durations (a few days) over a relatively small area. Horizontal transport associated with the Tibetan Plateau anticyclone caused the $CO_2$ mixing ratios to 300   vary between the UT and LS on a sub-continental scale.

*Data availability.* The TANSO–FTS data are available via the GOSAT Data Archive Service (GDAS) at https://data2.gosat.nies.go.jp/. The ECMWF Reanalysis version 5 (ERA5) data (daily basis) were taken from https://www.ecmwf.int/en/forecasts/datasets/reanalysis-datasets/era5.

*Author contributions.* Eguchi designed the study and prepared the manuscript, and Honda and Eguchi analysed the data. Saitoh developed 305   the $CO_2$ profile data. All authors discussed the data and manuscript.

*Competing interests.* The authors declare that they have no conflicts of interest.

*Acknowledgements.* This study was undertaken within the framework of the GOSAT Research Announcement and conducted as a joint research program of CEReS, Chiba University (2019–2021). This research was partially supported by a Grant-in-Aid for the 2019–2021 Initiative for Realising Diversity in the Research Environment through the Diversity and Super Global Training Program for Female and 310   Young Faculty (SENTAN-Q).



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





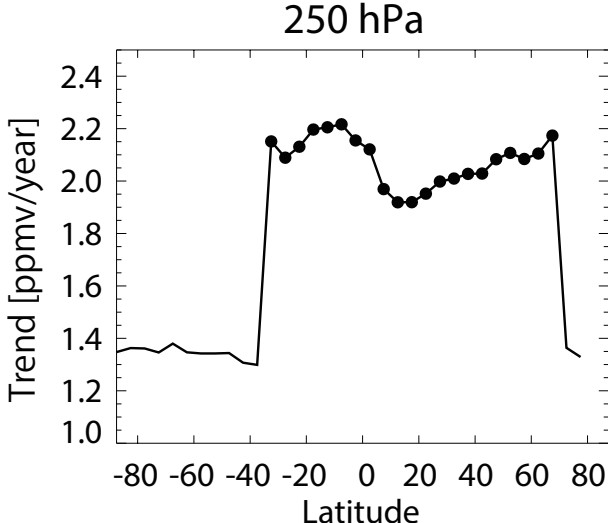

**Figure 1.** Latitudinal distribution of the trend of the $CO_2$ mixing ratio at 250 hPa from January 2010 to December 2013. The dotted points indicate the latitude where the bias correction was applied ($30°S-70°N$).

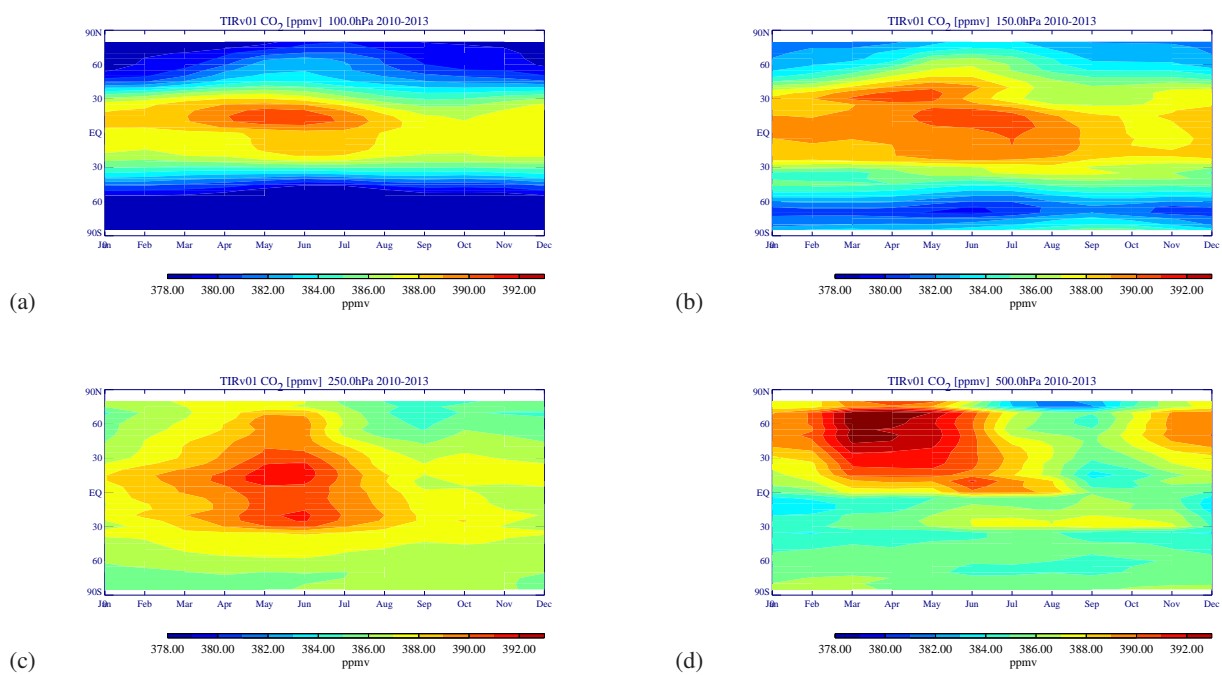

**Figure 2.** Time and latitude section of $CO_2$ mixing ratios averaged over four years from January 2010 to December 2013 at (a) 100 hPa, (b) 150 hPa, (c) 250 hPa, and (d) 500 hPa.

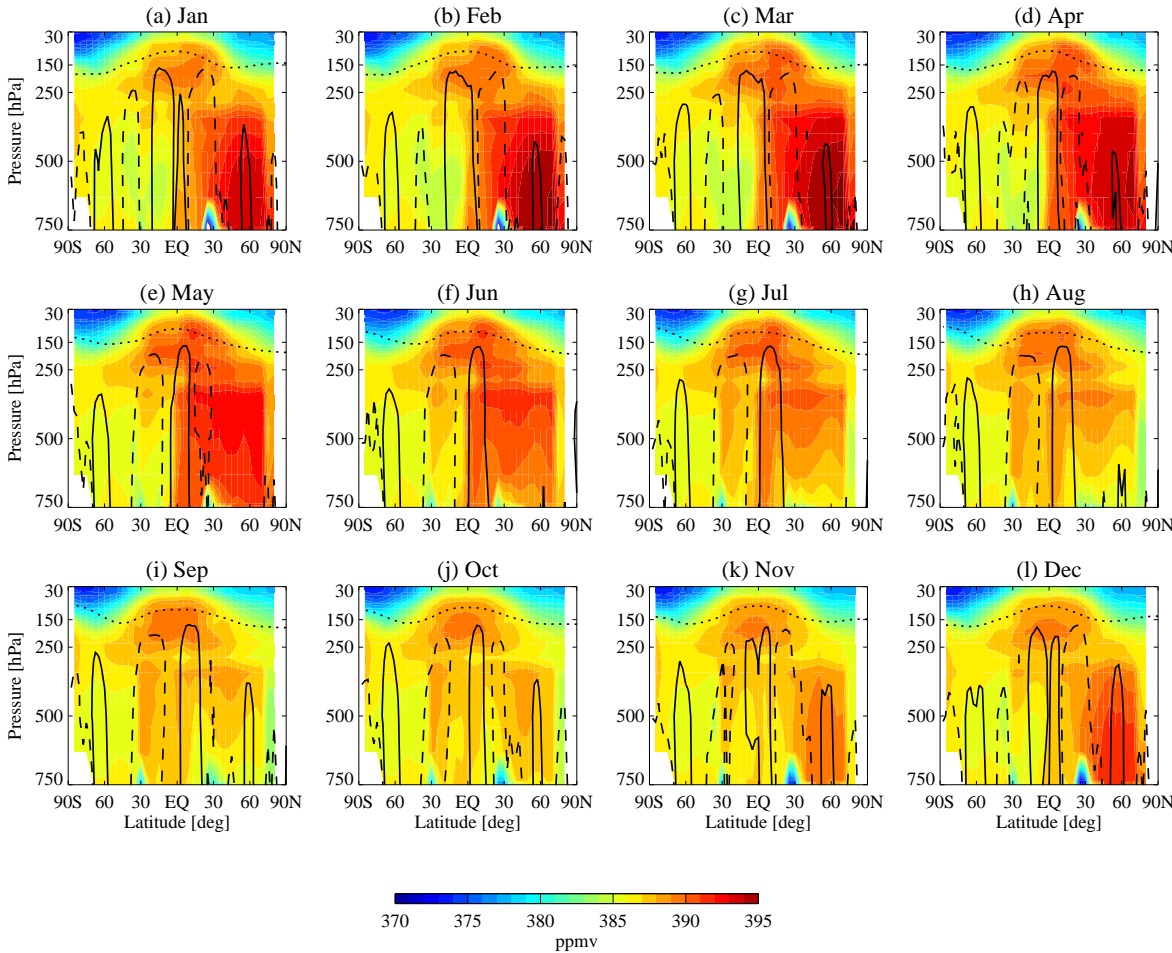

**Figure 3.** Latitude–pressure section of $CO_2$ mixing ratios averaged over four years from 2010 to 2013 for each month, with de-trending applied. Dotted contours show potential temperatures of 370 K, and solid and dashed contours show negative and positive pressure vertical velocities (–1.0 and 1.0 Pa/s), respectively.





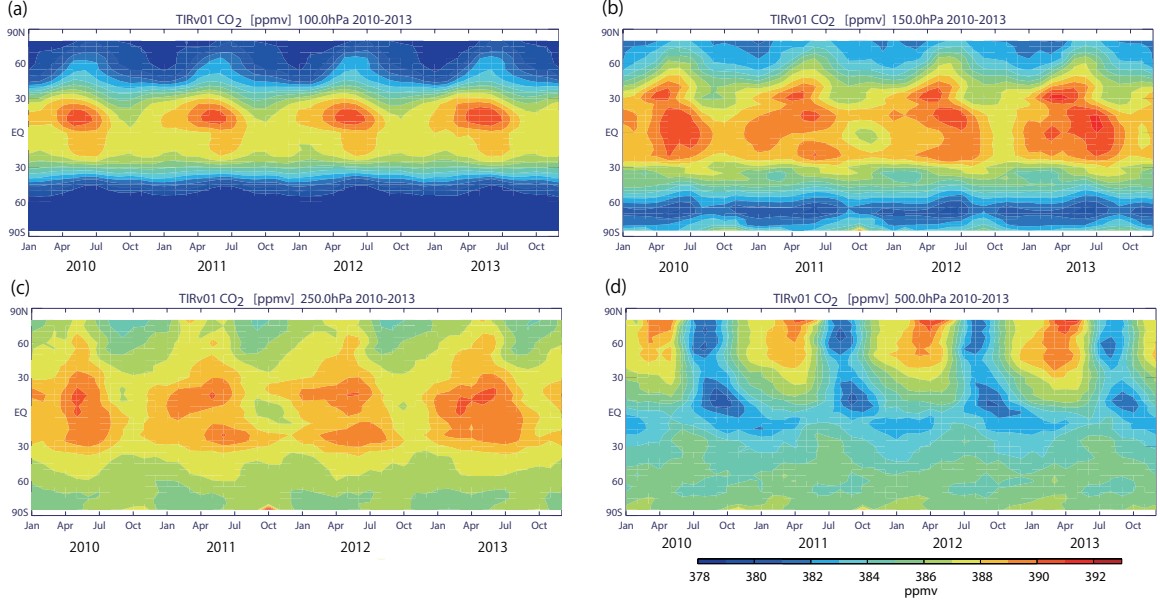

**Figure 4.** Time and latitude section of $CO_2$ mixing ratios from January 2010 to December 2013, without trends at each latitude and pressure, for (a) 100 hPa, (b) 150 hPa, (c) 250 hPa, and (d) 500 hPa.

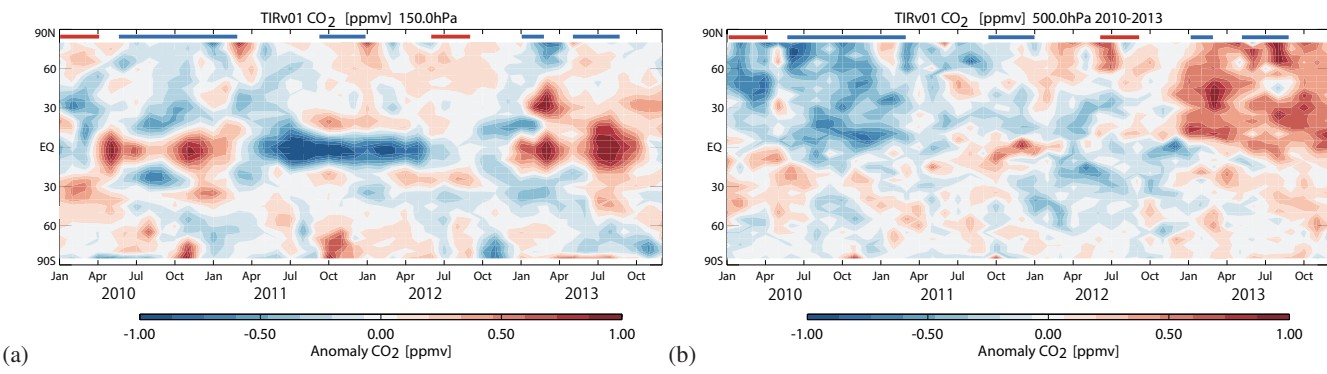

**Figure 5.** Same as Figure4, but for the anomaly from the monthly average. (a) 150 hPa and (b) 500 hPa. The red (blue) horizontal lines at the top of panels show the sea surface temperature in Nino3 (5°N–5°S, 150°W–190°W) higher(lower) than 0.5 degree C from the 30-year average over there.



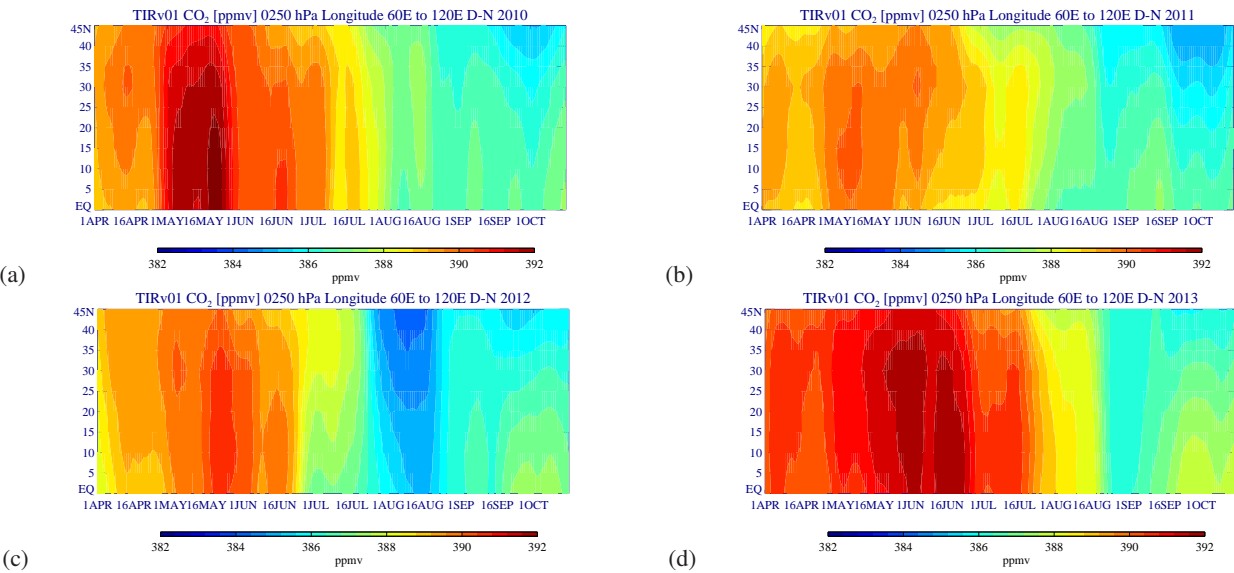

**Figure 6.** Time and latitude section of $CO_2$ mixing ratios at 250 hPa in each year averaged between $60°$E and $120°$E and shown as a 5-day running mean without trends at each latitude and pressure.

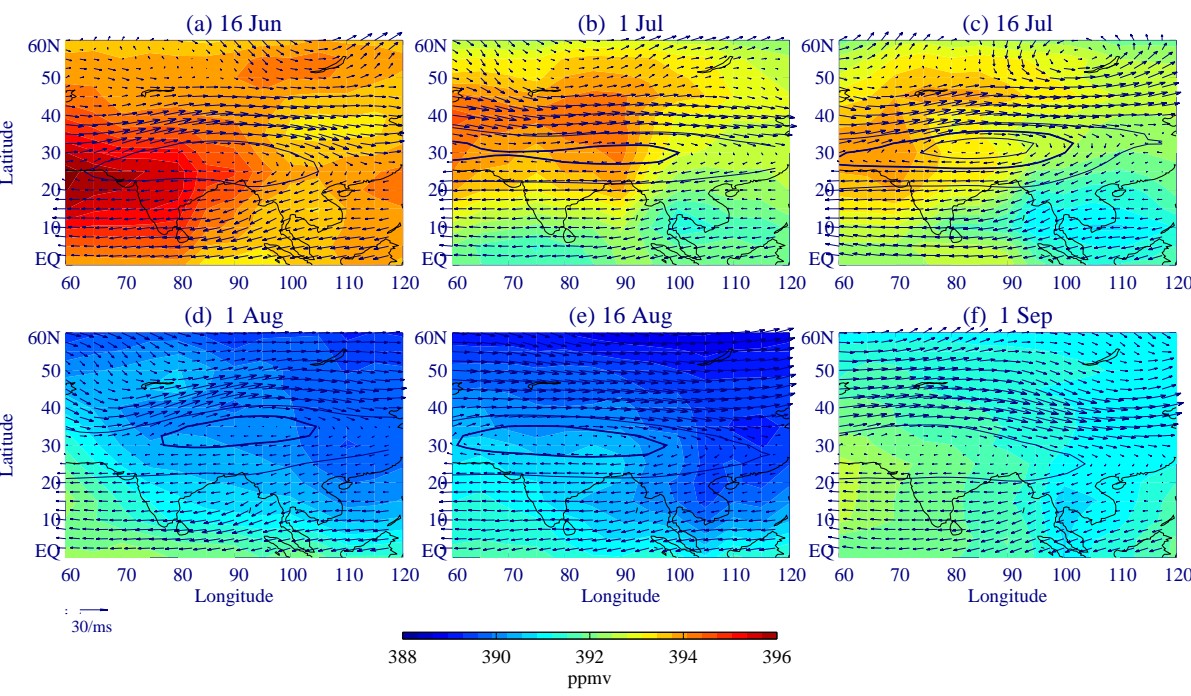

**Figure 7.** Map of $CO_2$ mixing ratios at 250 hPa as a 14-day average from (a) 16 June 2012, which shows the horizontal winds [m/s] (vectors) and geopotential height [$m^2/s^2$] (contours). Contour lines represent 11000, 11040, and 11060 [m]. (b–f) The thick contour is 11040 [m]. The same as (a), but for the 14-day averages from 1, July, 16 July, 1 August, 16 August, and 1 September, respectively.





**Table 1.** The trend [ppmv year$^{-1}$] (average, standard deviation, and minimum and maximum values) at each pressure level derived from GOSAT TANSO–FTS TIR data. The trend was calculated for the latitude range of 80°S to 80°N at 100 and 150 hPa and 30°S to 70°N at 250 and 500 hPa, at 5° intervals (January 2010 to December 2013).

| Trend from 2010 to 2013 [ppmv year$^{-1}$] | Pressure [hPa] | | | |
|---|---|---|---|---|
| | 100 | 150 | 250 | 500 |
| Ave. | 2.02 | 1.82 | 2.08 | 1.87 |
| STD | 0.08 | 0.31 | 0.09 | 0.30 |
| Max. | 2.15 | 2.17 | 2.22 | 2.35 |
| Min. | 1.86 | 1.22 | 1.92 | 1.50 |