# Peer review of "Carbon dioxide variations in the upper troposphere and lower stratosphere from GOSAT TANSO–FTS TIR profile data"

_Atmospheric Chemistry and Physics, 2022_

## Referee Comment (RC1)

Review to "Carbon dioxide variations in the upper troposphere and lower stratosphere from GOSAT TANSO-FTS TIR profile data" (acp-2022-46)

This study investigated the CO2 mixing ratios in the UT–LS region and analyzed the CO2 trend, seasonal and intra-seasonal variation and its link to large scale circulation, ENSO and Asian summer monsoon etc. It is no doubt that such a study to the trend and variation of CO2 in UT-LS region is very important for global warming study, and our knowledge in this topic is limited due to the limited observations. Therefore, using GOSAT-FTS retrieval products is a novel approach. The method used is straightforward. With some work, it can become a good paper to publish in this journal.

**Major problem:**

I doubt if the first author fully realizes the limitation of satellite observation using TIR channels. The major sensitivity of TIR sensor is in the middle to upper troposphere, and its sensitivity is less in the LS and LT region. So, my major concern is: if or how much of the derived trend and seasonal variation is from the a-priori used. If it is largely from the a-priori, like using model data as a-priori that include similar trend and seasonal variation, the derived trend and seasonal variation could be wrong. Therefore, I think it is needed to add in the context: the a-priori used in the retrieval, the DOFs used to screen the data. The uncertainty/range of the derived CO2 increase rate should be added.

**Minor problem:**

The presentation is overall pretty good, but some sentences need to revise. Below is a list of some of them, and I would encourage the authors to go through the whole manuscript.

L26: derived ?

L67: but about 80,000 observations of the same quality? it is an overstatement. Need to revise

L98: the atmospheric mixing ratio data are of high accuracy? it is also an overstatement as L67.

"the accuracy is about 4 ppmv for XCO2 " has nothing to do with GOSAT-FTS.

L100 - L104: What about night-time GOSAT-FTS products and its use ? how to do quality control without CAI?

L109: The present study defined the UT–LS region as 287.30–90.85 hPa. This is inconsistent with that in abstract and other places (maybe). Please check.

L112: The algorithm has a sensitivity peak in the upper troposphere? No matter how smart an algorithm you can design, the sensor itself is the key.

L120-121: ... less than 0.5 ppmv at lower latitudes and 1.0 ppmv at middle and high latitudes? It is confused what are you talking about.

L124: conducted a bias correction at "each month"? Should not be that way.

L142: revise this sentence

L145: citation is missing

L166: this  $\rightarrow$  it

L184-185: The 370 K potential temperature defines the physical surface of the tropopause. Where is this from ? also PV=2 PVU is better to use to define the dynamic tropopause.

L218-219 "The minimum CO2 mixing ratios were located at latitudes higher than  $60^{\circ}$ N and at around  $15^{\circ}$ N from June to November. ". It is confused and needs to revise.

Figure 5: caption. Need to revise.

About the link with ENSO, you can check the following paper

Corbett, A., X. Jiang, X. Xiong, A. Kao, and L. Li, 2017, Modulation of midtropospheric methane by El Niño, Earth and Space Science, 4, doi:10.1002/2017EA000281.

---

## Author Comment (AC1)

Dear Dr. Xiong as Reviewer#1

Thank you very much for reviewing our manuscript and giving us many useful comments.
We agree your almost all of comments and suggestions, therefore the manuscript has been revised along your (and another reviewer) comments. Please check the following our reply(black) of each your comments (blue).
In addition, we are sorry for late replaying because it takes time to repair the data server which was broken down.

The major points that we deal with in the revised manuscript are as follows:
1. Figures were updated or removed.
   ① Adding 2PVU (potential velocity unit) and OLR (outgoing longwave radiation) as the indicator of deep convection to Figures 1, 2, 3 and 5 of the revised manuscript, and zonal wind to Figure 2 of the revised manuscript
   ② Figure 1 (plot of latitudinal distribution of increasing rate) in the previous manuscript was removed because Table 1 was enough information on it.
   ③ Figure 3d (time-latitude cross section for 4-year at 250 and 500hPa) of the revised manuscript was drawn by the corrected data, because the previous figure was drawn by the data without the bias correction.
   ④ Figure 5 (time-latitude cross section of inter-annual variation) in the previous manuscript was removed from the revised manuscript.
2. The discussion on $CO_2$ variation related with ENSO was moved to section of Discussion and Summary. Therefore the section name was changed to "Summary and Conclusion" to "Discussion and Summary" in the revised manuscript.
3. Many references recommended by reviewers were added to mainly the section of Introduction related with the *in situ* observational studies.

Review to "Carbon dioxide variations in the upper troposphere and lower stratosphere from GOSAT TANSO-FTS TIR profile data" (acp-2022-46)

This study investigated the CO2 mixing ratios in the UT–LS region and analyzed the CO2 trend, seasonal and intra-seasonal variation and its link to large scale circulation, ENSO and Asian summer monsoon etc. It is no doubt that such a study to the trend and variation of CO2 in UT-LS region is very important for global warming study, and our knowledge in this topic is limited due to the limited observations. Therefore, using GOSAT-FTS retrieval products is a novel approach. The method used is straightforward. With some work, it can become a good paper to publish in this journal.

Major problem:

I doubt if the first author fully realizes the limitation of satellite observation using TIR channels. The major sensitivity of TIR sensor is in the middle to upper troposphere, and its sensitivity is less in the LS and LT region. So, my major concern is: if or how much of the derived trend and seasonal variation is from the a-priori used. If it is largely from the a-priori, like using model data as a-priori that include similar trend and seasonal variation, the derived trend and seasonal variation could be wrong. Therefore, I think it is needed to add in the context: the a-priori used in the retrieval, the DOFs used to screen the data. The uncertainty/range of the derived CO2 increase rate should be added.

A: Thank you for the useful comment. Along your comments on a priori and DOF, the following

sentences were added to 2.2 sub-section of the revised manuscript. P.5, line 140-146

*"In addition, data with a higher degree of freedom (DOF), with more than two times the standard deviation from the average of each month, were used for the analysis, because we intended to use data that were not constrained by a priori information, which is taken from the NIES transport model version 5 (NIES-TM05) (Saeki et al., 2013). The retrieved $CO_2$ data at UT fit with COTNRAIL rather than a priori data. Furthermore, the retrieved data allow the concentrations in the UT and LS to be distinguished (see figure 6b of Saitoh et al. (2016)). The magnitude of DOF highly depends on latitudes, high at low latitudes (2.25), and high latitudes (1.25) in both hemispheres., There is little seasonal change in the DOF data. The percentage of data screened by the DOF was about 2%-5%."*

On the uncertainty / range of increase rate, the standard deviation of increasing rate was shown in Table 1. The Figure 1 shown in the previous manuscript was removed because Table 1 covers the information of increasing rate at UTLS. On the other hand, the bias correction was done with considering the inter-annual variation at each latitudinal band and season by using CONTRAIL data as shown in Saitoh et al (2017).

Minor problem:

The presentation is overall pretty good, but some sentences need to revise. Below is a list of some of them, and I would encourage the authors to go through the whole manuscript.

L26: derived ?
A: corrected at p.2, l.25

L67: but about 80,000 observations of the same quality? it is an overstatement. Need to revise
A: The term "of the same quality" was removed, the sentence was revised as follow.
*"about 80,000 observations, both with and without cloud cover"* at p. 3, l. 80

L98: the atmospheric mixing ratio data are of high accuracy? it is also an overstatement as L67.
A: This sentence was removed because the description of $XCO_2$ was not needed for $CO_2$ profile data.

"the accuracy is about 4 ppmv for XCO2 " has nothing to do with GOSAT-FTS.
A: We agree, this sentence was removed.

L100 - L104: What about night-time GOSAT-FTS products and its use ? how to do quality control without CAI?
A: Along the reviewer's comment, the explanation on cloud screening at the night-time are added to p.4, l.111-112.
*"The cloud screening at the night-time was based on the radiance around 900 cm-1 (around 11 μm) of the TANSO-FTS TIR band [Imasu et al., 2010]."*

The following reference was added to the list of reference section.
Imasu, R., Y. Hayashi, A. Inagoya, N. Saitoh, and K. Shiomi: Retrieval of minor constituents from thermal infrared spectra observed by GOSAT TANSO-FTS sensor, P. Soc. Photo-Opt. Inst, 7857, 785708, 2010.

L109: The present study defined the UT–LS region as 287.30–90.85 hPa. This is inconsistent with that in abstract and other places (maybe). Please check.

A: The UTLS is generally defined the region, 300 – 70 hPa. In this study, we use only altitudes that can correct bias using the results of comparison with aircraft observations in previous studies. In addition, along the other reviewer's suggestion the explanation of the averaging kernel was added. The text was modified as follows. P.4, l.121- p.5, l.125.

*"The averaging kernel in Figure 1 of Saitoh et al. (2016) shows the sensitivity of UT, particularly at 300 -- 200hPa at lower and middle latitudes. The present study used the level between 287.30 and 90.85 hPa as the UT–LS region. The numbers of retrieved layers vary from 9 to 14 (see table 1 of Saitoh et al. (2016)), which yield lower (upper) pressure levels of 287.30 (237.14), 237.14 (195.73), 195.73 (161.56), 161.56 (133.35), 133.35 (110.07), and 110.07 (90.85) hPa, respectively."*

L112: The algorithm has a sensitivity peak in the upper troposphere? No matter how smart an algorithm you can design, the sensor itself is the key.
A:  As same answer as above, as shown in Figure 1(a,b) of Saitoh et al. (2016), the TIR band has sensitive at UT, especially between 200 and 300 hPa, in low- and middle latitudes,.

[Figure]

Figure S1 : Averaging Kernel functions at lower latitude in summer (a) and middle latitude in spring of TIR band, adapted from Figure 1 (a,b) of Saitoh et al. (2016).

The sentence was modified as above.

L120-121: … less than 0.5 ppmv at lower latitudes and 1.0 ppmv at middle and high latitudes? It is confused what are you talking about.

A: The sentence was modified as follow, p. 5, l.133-135.

"\Saito et al. (2017) found that the bias of $CO_2$ data as compared with CME observations at approximately 11 km was less than 0.5 ppmv at low latitude and 1.0 ppmv at mid- and high latitudes."

L124: conducted a bias correction at "each month"? Should not be that way.
A: The term was revised from "each month" to "each season". P.5, l.138.

L142: revise this sentence
A: the sentence was revised as follow

"For analysis of the inter-annual $CO_2$ variations, we used the anomaly data from the four-year monthly mean derived from equation (2), after subtracting the **seasonality** for each month and pressure and latitude." , p.6, l.158-160

L145: citation is missing
A: Add the following reference to p.6, l.162 and the list of reference of the revised manuscript.

Hersbach, H., Bell, B., Berrisford, P., Hirahara, S., Horanyi, A., Munoz-Sabater, J., Nicolas, J., Peubey, C., Radu, R., Schepers, D., Simmons, A., Soci, C., Abdalla, S., Abellan, X., Balsamo, G., Bechtold, P., Biavati, G., Bidlot, J., Bonavita, M., De Chiara, G., Dahlgren, P., Dee, D., Diamantakis, M., Dragani, R., Flemming, J., Forbes, R., Fuentes, M., Geer, A., Haimberger, L., Healy, S., Hogan, R. J., Holm, E., Janiskova, M., Keeley, S., Laloyaux, P., Lopez, P., Lupu, C., Radnoti, G., de Rosnay, P., Rozum, I., Vamborg, F., Villaume, S., and Thepaut, J. N.: The ERA5 global reanalysis, Q. J. R. Meteorol. Soc., 146, 1999–2049, 2020.

L166: this → it
A: corrected, p.6. l.183

L184-185: The 370 K potential temperature defines the physical surface of the tropopause. Where is this from ? also PV= 2 PVU is better to use to define the dynamic tropopause.
A: The revised manuscript uses the 2 PVU for physical tropopause and add the potential temperature of 380K along the other reviewer suggestion.

L218-219 "The minimum $CO_2$ mixing ratios were located at latitudes higher than 60°N and at around 15°N from June to November. ". It is confused and needs to revise.
A: The sentence was revised as follow,
"The minimum $CO_2$ mixing rations were located at higher latitudes than 60N from June to October in the northern hemisphere and around 15 N from July to November.", p.8. l.246-247.

Figure 5: caption. Need to revise.
A:This figure and its related description were removed because the other reviewer's comment pointed out that the speculative sentences and diagrams especially ENSO. The discussion related

with ENSO was moved to the section of "Discussion and Summary".

About the link with ENSO, you can check the following paper
Corbett, A., X. Jiang, X. Xiong, A. Kao, and L. Li, 2017, Modulation of midtropospheric methane by El Niño, Earth and Space Science, 4, doi:10.1002/ 2017EA000281.
A: Thank you for giving the useful paper. In the revised manuscript the description on ENSO was moved to the section of "Discussion and summary". The suggested reference is added to p.11, l.323-325.

---

## Author Comment (AC2)

Dear Reviewer#2

We would like to thank you very much for reviewing our manuscript carefully and giving us many useful comments.

We agree your almost all of comments and suggestions, therefore the manuscript has been revised along your (and another reviewer's) comments. Please check the following our reply (black) of each your comments (blue).

In addition, we are sorry for late replying because it takes time to repair the data server which was broken down.

The major points that we deal with in the revised manuscript are as follows:

- 1. Figures were updated or removed.
  - ① Adding 2PVU (potential velocity unit) and OLR (outgoing longwave radiation) as the indicator of deep convection to Figures 1, 2, 3 and 5 of the revised manuscript, and zonal wind to Figure 2 of the revised manuscript
  - ② Figure 1 (plot of latitudinal distribution of increasing rate) in the previous manuscript was removed because Table 1 was enough information on it.
  - ③ Figure 3d (time-latitude cross section for 4-year at 250 and 500hPa) of the revised manuscript was drawn by the corrected data, because the previous figure was drawn by the data without the bias correction.
  - ④ Figure 5 (time-latitude cross section of inter-annual variation) in the previous manuscript was removed from the revised manuscript.
- 2. The discussion on CO2 variation related with ENSO was moved to section of Discussion and Summary. Therefore the section name was changed to "Summary and Conclusion" to "Discussion and Summary" in the revised manuscript.
- 3. Many references recommended by reviewers were added to mainly the section of Introduction related with the *in situ* observational studies.

**Review of Honda et al.,**

First of all apologies for my late comment, which was due to unforeseeable issues. However I'm sorry to say, that I had a hard time reading the manuscript. The data set is very interesting and clearly of interest to the community. The discussion of the link to transport and dynamics is however misleading or partly wrong and neglects many aspects of transport (e.g., the role of the tropopause as a barrier for transport and its effect on the CO2 cycle amplitude or phase, which has already been discussed in several papers).

The paper presents observations of CO2 from GOSAT from 2010 to 2013, which is analyzed at pressure levels of 500, 250, 150, and 100 hPa. The authors detrend the data by applying a simple linear empirical fit to build multiannual climatologies and anomalies. They show the zonal mean distributions of CO2 and at the levels mentioned avbove to conclude on the transport processes, which cause the observed CO2 distributions, but neglect relevant literature (Boering, Andrews, Sawa, Hoor, ...).

For the troposphere they state a transport time of two months from the LT (lower troposphere) to the UT (upper troposphere) and a dampening of 50% of the seasonal cycle amplitude. They relate this to the "absolute mixing ratio decreasing with altitude and to a lesser extent mixing with low CO2 mixing ratio air mass".

Further they link troipcal CO2 mixing ratios to ENSO and identify interannual variability of the CO2 in the monsoon region to different relations of vertical transport by convection and

horizontal transport "via anticyclonic circulation". The authors neglect relevant literature of CO2 and its seasonal cycle from aircraft observations.

A: Thank you again for useful comments, especially providing us many related references. The results of previous studies have been added to the introduction section (the 3rd paragraph in page 2.) and proper location of the revised manuscript.

Figure 5 and related paragraphs with ENSO has been moved to "Discussion and Summary" section because, as you point out, long-term variability has many factors and has become presumptive discussion, so we decided to focus on the climatological descriptions of CO2 at the revised manuscript.

The paper does not include any analysis of transport via e.g. Lagrangian methods, nor it shows links to surface observations or at least comparisons to the interannual variations of emissions or surface distributions or variability (zonally, globally or regionally, e.g. monsoon).

A: We agree that the discussion of UTLS  $CO_2$  variation is needed to transport. As you pointed out, we added a description of the contribution of vertical and horizontal transport, and detailed descriptions of vertical and horizontal transport in the chapter on intra-seasonal variability with a short time scale (sub-section 3.4).

On the other hand, even though the results are very similar to ground observations, the upper air masses are the result of atmospheric transportation processes and mixing with other air masses. Therefore it is difficult to discuss UTLS CO2 fluctuations from numerical calculations that cause chaotic errors, especially longer integration time causes larger error.

The authors further discuss transport and mixing particular at 250hPa, but do not even mention the term "subtropical jet", mixing barrier, isentropic transport, and consequently do not discuss their roles for the propagation of the seasonal cycle. The also state that Theta=370 K "indicates the physical surface of the tropopause", which is simply wrong. They fully dismiss the role of the extratropical tropopause as transport barrier, when discussing the timing of the seasonal cycle and its amplitude change at the barrier.

A: We thank for making us aware of above points. Along your recommendation, the jet and physical surface '380 K' were added to figures, and the related description were added to the result section.

They state, that the role of seasonal CO2-cycles has not been studied and neglect significant corresponding work: For the stratosphere above 100 hPa: Andrews et al., 1999, Boering et al., 1994, 1996, Strahan et al., 1998.

For the UTLS and lower stratosphere: Hoor et al., 2004, Engel et al., 2007, Boenisch et al., 2009. For upper troposphere and the monsoon: Schuck et al., 2010, Gurk et al., 2008.

A: Also many thank you for giving us these information. Those previous papers were added to the introduction and its suitable sections.

All in all there are too many speculations when linking transport and CO2 observations. I recommend to resubmit it focusing on the climatologies and the observations and carefully linking them to e.g. surface seasonal cycles from global observational network for the LT/MT data. For the UTLS there must be a correct treatment of the tropopause particularly for the 150 hPa and 250 hPa level. One could e.g. derive distinct seasonal cycles for tropospheric and

stratospheric data, which can be compared to existing data sets (see references). Speculations about transport mechanisms should be removed.

A: Thank you for your very reasonable opinion.

The speculative sentences and diagrams were removed and other variables (wind field and OLR as indicator of convection) which make it easier to understand the transportation were added.

Therefore I can't recommend the paper for publication in the current form.

I do highly suggest a resubmission with a different focus, since the data set as such is very valuable, but the discussion of potential links to transport and mixing is inappropriate. I encourage the authors for resubmission either sharpening the transport discussion or just focusing on the climatologies.

A: We agree the reviewer's comment, we said again that the speculative sentences and diagrams were removed, especially the inter-annual variation related with ENSO was moved to the section of "Discussion and Summary".

**Minor points: line 109: What is the vertical resolution and how do averaging kernels look like?**

A: The information of vertical resolution and the averaging kernel was added to revised manuscript, P.4, 1.121- p.5, 1.125.

The averaging kernel in Figure 1 of Saitoh et al. (2016) shows the sensitivity of UT, particularly at 300 -- 200hPa at lower and middle latitudes. The present study used the level between 287.30 and 90.85 hPa as the UT–LS region. The numbers of retrieved layers vary from 9 to 14 (see table 1 of Saitoh et al. (2016)), which yield lower (upper) pressure levels of 287.30 (237.14), 237.14 (195.73), 195.73 (161.56), 161.56 (133.35), 133.35 (110.07), and 110.07 (90.85) hPa, respectively."

Fig. 1: Gradients appear at the tropopause. These were not accounted for. The discussion of trends and Figure 1 illustrates an example of the coarse and insufficient discussions and speculations: The trend figure is discussed without any mentioning of the tropopause and its role for e.g. the mid-lat trend. The according table 1 provides trends for different latitude ranges, different altitudes without consideration or discussion of the tropopause. What shall one learn from this?

A: Figure 1 was removed from the revised manuscript because the increasing rate are found by Table 1.

Fig.6 and related discussion (lines 261-268): The monsoon plays of course a role for the observed 250 hPa CO2 in Fig.6, but there is no disussion of potential surface emission variations, change of ENSO-related tropospheric circulation pattrens, change of emission patterns, the authors state without any supporting analysis, that the observed CO2 variability is related to variability of deep convection. How do the authors come to their conclusion? How is emission variability differentiated from large scale transport variability or convection? A: The description on ENSO was removed from the revised manuscript and was moved to discussion section because there are many factors that affect long-term fluctuations, and it is not possible to discuss all of them in this study.

Fig.2 : Concerning the bias correction, which is also mentioned in the manuscript: Which role does the isentropic CO2 gradient at the extratropcal and subtropical tropopause play for the

bias correction? Did the authors consider the tropopause when calculating the bias?

A: The isentropic surface (380K potential temperature) and PVU=2 line were added to the figures (Figures 1,2,3, and 5) of the revised manuscript. With the revision of figure, the description of Figure 1 was modified in Sub-section 3.2, for example, line 190-192 in p.7. This revised figure is easier to understand the horizontal and vertical transports, for example at 250 hPa, the CO2 maximum at the lower latitude vary with vertical transport, on the other hand it is suggested that the extension of CO2 higher mixing ration to the higher latitudes at northern hemisphere across +2PVU line was affected by the horizontal transport.

Fig 2d) How do the cycles (e.g. at point Barrow and Mauna Loa) fit to the GOSAT observations at 500 hPa. Highest CO2 at high latitudes should occur later than at low latitudes, since biological activity is delayed. How does this fit to Fig.2d? also line 165-168.

A: The following figure (FigureA) shows the  $CO_2$  variations at MLO site, the GOSAT TIR  $CO_2$  data was taken +/-5 degrees around MLO site. At 500hPa, the seasonal changes from GOSAT TIR are almost the same, and the amplitude decreases with increasing altitude, and the maximum peak timing is delayed by about one month. It was found that the UTLS  $CO_2$  derived from GOSAT TIR shows the similar results with the previous studies.

Figure S2: Time series of CO2 at MLO. The black line shows the surface observation at MLO, the red, light blue and purple lines show the GOSAT observation averaged  $\pm$  5 degree box around MLO at 150, 250 and 500 hPa, respectively.

1.184: The 370K isentrope defines the physical surface of the tropopause. This statement is simply wrong. The tropopause is no way defined by isentropes (read Holton, 1995, Hoskins, 1991, Bethan, 1996...

A: The isentropic surface was modified as 380K potential temperature through the revised manuscript.

Fig. 3: How well is the troposphere resolved in the CO2-data (vertical resolution, kernels, degrees of freedom)?

A: From the Table 1 and Figure 1 of Saitoh et al (2016), the vertical resolution and the averaging kernel (AK) were enough to discuss the vertical transport. The AK explanation was added to the revised manuscript, p.4, 1.121-122.

'Caption': Replace 'vertical velocity' with 'pressure tendency' – they have different signs.

A: The term was modified, p19.

Fig.4b)c): The data at 250 hPa are affected by the tropopause location, which inhibits quasihorizontal (quasi-isentropic) transport. The phase propagation therefore is different from 100 hPa or 500 hPa (see Sawa et al., 2008, Hoor et al., 2004). Please add the (mean) 2 PVU and 4 PVU contour (also Fig. 6a)-d) and Fig.5 a)

A: The 2 PVU line was added to figures. With the revised figure, the description was modified in sub-section 3.3, p.8-9.

1.208/209: Which vertical gradient? Please calculate or plot (e.g. for diffenet latitudes).

A: The following figure (FigureS3) shows the latitudinal distribution of the vertical gradient between 250hPa and UT/LS pressure levels. It was found that the latitudinal gradient between 10S and 30S in the southern hemisphere was steeper than that in northern hemisphere (10N-30N), and the temporal variation of latitudinal gradient in northern hemisphere was larger than that in southern hemisphere at each pressure levels.

The following sentence was added to the revised manuscript, p.8, 1.235-239.

"The latitudinal gradient between 250 hPa and 100 hPa in the Southern Hemisphere was steeper than in the Northern Hemisphere and, for example, the CO2 mixing ratio decreased 1.45 ppmv from 10S to 30S and 0.71 ppmv from 10N to 30N in annual mean. However the latitudinal gradient in the NH varies seasonal (approximately 2.8 ppmv); the gradient in July (January) was steeper (gentler), on the other hand the seasonal variation of gradient in SH was smaller (approximately 0.9 ppmv)."

Figure S3: The latitudinal distribution of vertical gradient of  $CO_2$  [ppmv] between 237hPa and 90 hPa (a), 110 hPa (b) and 133 hPa (c). The green lines show each month except January (black) and July (blue), the red line shows annual mean. The values in each panels indicate the maximum (top), average (red) and minimum (bottom) of latitudinal gradient between 10S(N) and 30S(N) at left (right) side of each plot.

1.210: This statement holds for any tracer and is very unspecific - the distribution of anything in the UTLS depends on vertical and horizontal transport in the troposphere. A: This sentence was removed to avoid the confusion.

Fig.5: 500 hPa shows a trend of the anomaly at higher latitudes. Is this due to the (possibly wrong) linear trend estimate to derive the anomaly (eqn.1)?

A: Figure 5 was removed from the revised manuscript.

Also Figure 7: Why is the CO2 maximum related to deep concvection? Why do the data not

show any accumulation effect inside the anticyclone (see e.g. Baker, 2013, Schuck, 2010)?

A: The figure (Figure 6 in the revised manuscript) was modified. The revised figure shows the high mixing ration was located within the Asian monsoon high at the June and July. And the explanation of this figure was written more carefully in Section 3.4, at 1.290-303 in p.10.